# Clinical Significance of Gray to White Matter Ratio after Cardiopulmonary Resuscitation in Children

**DOI:** 10.3390/children9010036

**Published:** 2022-01-01

**Authors:** Yun-Young Lee, Insu Choi, Seung-Jae Lee, In-Seok Jeong, Young-Ok Kim, Young-Jong Woo, Hwa-Jin Cho

**Affiliations:** 1Department of Radiology, Chonnam National University Children’s Hospital, Gwangju 61706, Korea; yunyoung0219@gmail.com; 2Department of Pediatrics, Chonnam National University Children’s Hospital and Medical School, Gwangju 61706, Korea; realalice@hanmail.net (I.C.); ik052@jnu.ac.kr (Y.-O.K.); yjwoo@jnu.ac.kr (Y.-J.W.); 3Department of Pediatrics, KS Hospital, Gwangju 62248, Korea; runningami@gmail.com; 4Department of Thoracic and Cardiovascular Surgery, Chonnam National University Hospital and Medical School, Gwangju 61706, Korea; isjeong1201@gmail.com

**Keywords:** cardiopulmonary resuscitation (CPR), pediatric, survival, neurologic outcome, gray to white matter ratio (GWR)

## Abstract

Cardiopulmonary resuscitation (CPR) successfully restores systemic circulation approximately 50% of the time; however, many successfully restored patients have severe neurologic damage. In adults, the gray matter to white matter attenuation ratio (GWR) in brain computed tomography (CT) correlates with the neurologic outcome. However, in children, the clinical significance of GWR still remains unclear. The aim of this study was to evaluate the clinical characteristics of children who underwent CPR for cardiac arrest according to the survival and to demonstrate the differentiation of grey/white matter by Hounsfield units of brain CT and to characterize the attenuations of grey and white matters. Methods: This is a retrospective single-center study. We enrolled those who underwent brain CT within 24 h after return of spontaneous circulation (ROSC) from January 2005 to June 2018. Brain CTs were taken within 24 h of ROSC. We measured the attenuation of grey and white matter in Hounsfield units and calculated GWR. They were compared with healthy controls. Patients were analyzed as follows: survivors vs. non-survivors and better neurologic outcome vs. worse neurologic outcome. Results: Among 100 pediatric patients who had CPR, 56 met inclusion criteria. There were 24 patients who survived and 32 non-survivors. Our study revealed that the incidence of seizure, duration of CPR, and instances of hypothermia were significantly different between survivors and non-survivors. In both survivors and non-survivors, the attenuation of the caudate nucleus, putamen, GWR-basal ganglia, and average GWR were significantly different from controls. In regression analyses, the medial cortex and average GWR were the significant variables to predict survival, and the receiver operating curves revealed areas under curve of 0.733 and 0.666, respectively. Also, the medial cortex 1 was the only variable that predicted the neurologic outcome. Conclusions: There was some predictive survival value of GWR and medial cortex at the centrum semiovale level in early brain CT within 24 h after cardiac arrest. Although we could not find the predictive value of GWR in the neurologic outcome of pediatric patients, we found that the absolute attenuation of the medial cortex was low in patients with worse neurologic outcomes. Further prospective, multicenter studies are needed to determine the predictive value of GWR and the medial cortex.

## 1. Introduction

Although cardiopulmonary resuscitation (CPR) is successful in approximately 50% of patients for the restoration of systemic circulation (ROSC) [1], more than half of resuscitated patients remain comatose. This frequently leads to profound long-term neurologic sequelae or death in children after cardiac arrest [2].

To predict the survival and neurologic outcome in adults, electroencephalography (EEG), light reflex, somatosensory evoked potential, and brain computed tomography (CT) were previously evaluated [3]. In adults, gray matter to white matter attenuation ratio (GWR) in brain CT correlates with the neurologic outcome in patients with cardiac arrest, and it can be a useful objective early predictor of vegetative state or death in comatose patients after cardiac arrest [4,5].

Brain CT is one of the most frequently used diagnostic modalities that can be done immediately after stabilization of vital signs in children as well. There are several studies regarding GWR [6,7,8]; however, there are limited data of attenuation measurements of all grey and white matter in survivors and non-survivors.

The aim of this study was first to evaluate the clinical characteristics of children who underwent CPR for cardiac arrest according to their survival. Secondly, we aimed to demonstrate the attenuations of gray/white matters and GWR in early brain CT scans of children after cardiac arrest to explore the association of GWR with survival and neurological outcome.

## 2. Methods

We retrospectively reviewed the medical records of pediatric patients who had cardiac arrest unrelated to trauma and were resuscitated in the department of pediatrics of Chonnam National University Hospital in Republic of Korea between January 2005 and December 2018. Ethics approval for the study was obtained from the ethics committee of our university hospital (CNUH-2016-163), and the written consents were waived by the ethics committee.

We included patients under 18 years of age who satisfied the following criteria: survived for more than 24 h after ROSC; brain CT checked within 24 h of ROSC; no previous history of underlying neurological disease (e.g., epilepsy, hemorrhage, hydrocephalus). We collected demographic and clinical data of all patients, including age, gender, duration of CPR (min), the time taken initial CT after ROCS (hr), duration of intensive care unit (ICU) stay (day), primary cause of arrest, serum neuron-specific enolase (NSE), and serum lactate level. We further analyzed the characteristics of survivors and non-survivors. Survivors were defined as those who survived to hospital discharge.

Additionally, 50 healthy control patients under 18 years of age who underwent brain CT to evaluate their headache and had normal results were included.

Brain CT scans followed a standard protocol with a slice thickness of 5 mm, and regions of interest (ROI) were measured by a circular-shaped area (0.1 cm^2^) and average attenuation in Hounsfield units (HU) was recorded. For inter-rater agreement in statistics, ROIs were determined independently by two raters, and both were blinded to the clinical outcome (Figure 1). We checked the inter-observer variability and then used the mean of the value.

ROIs were assessed bilaterally in the caudate nucleus (CN), putamen (PU), forceps minor of the corpus callosum (CC), posterior limb of the internal capsule (PIC), thalamus (THL), medial cortex (MC), and medial white matter (MWM) at the level of the centrum semiovale and high convexity (MC1, MC2, MWM1, MWM2). GWR in basal ganglia (GWR_BG_) was calculated as follows: (CN + PU)/(CC + PIC). GWR in cerebrum (GWR_CE_) was calculated as follows: (MC1 + MC2)/(MWM1 + MWM2). Average GWR (GWR_AV_) was defined as the average of these 2 GWR (GWR_BG_ + GWR_CE_/2) [9].

Neurologic examination data of the patients were also reviewed, including Glasgow Coma Scale (GCS) score [10] and Glasgow Outcome Scale-Extended Pediatric Revision score (GOS-E Peds). GOS-E Peds was performed by pediatric neurologists on a scale in which 8 is the highest and 1 is the lowest (8 = Death; 7 = Vegetative State (VS); 6 = Lower Severe Disability (Lower SD); 5 = Upper Severe Disability (Upper SD); 4 = Lower Moderate Disability (Lower MD); 3 = Upper Moderate Disability (Upper MD); 2 = Lower Good Recovery (Lower GR); 1 = Upper Good Recovery (Upper GR)), with the following 7 subcategories to evaluate: Consciousness; Independence in the home; Independence outside the home; School/Work; Social & Leisure activity; Family & Friendships; Return to normal life [11].

The clinical characteristics and absolute attenuation in Hounsfield units of Brain CT were compared between survivors and non-survivors of cardiac arrest. The absolute attenuation of Hounsfield units and GWRs were compared among survivors, non-survivors, and healthy controls. The absolute attenuation of Hounsfield units and GWRs were compared according to the neurologic outcome.

### Statistical Analyses

Continuous variables are expressed as median and Interquartile range (IQR). The Mann–Whitney test was used to compare continuous variables between 2 groups (survivors vs. non-survivors), and the Kruskal–Wallis test for non-normally distributed data was used to compare continuous variables among 3 groups (survivors vs. non-survivors vs. healthy controls). A Chi-square test was used to compare non-continuous variables. For post-hoc analyses, the Conover method was used. In all analyses, *p*-values < 0.05 were considered to be statistically significant. Logistic regression was performed in those which were statistically significant. A receiver operating characteristic (ROC) curve analysis was performed. All analyses were performed using MedCalc Statistical Software ver. 19.1 (MedCalc Software bvba; Ostend, Belgium; http://www.medcalc.org; 2019: 1 December 2021).

## 3. Results

### 3.1. Patient Characteristics

From 2005 to 2018, 100 patients under 18 years of age were admitted to our emergency room for out-of-hospital cardiac arrest. Out of these patients, 44 were excluded, and 56 pediatric patients met inclusion criteria (Figure 2).

As described in Table 1, 38 patients were male, and 18 were female. There was no statistical difference between survivors and non-survivors. The median age was 5.5 years (IQR 0–13). Although there was not a significant difference between the ages of survivors and non-survivors, the age of non-survivors tends to be lower (median age 5.5 years vs. 2.5 years). The median age of controls were 10.5 (IQR 3.5–14.0). 

The primary causes of arrest were cardiac in 13 patients (23.2%), respiratory in 28 patients (50%), and others in 15 patients (26.8%). Others include near-drowning (*n* = 7), sudden infant death syndrome (*n* = 5), hanging (*n* = 1), vomiting, diarrhea, and hypovolemic shock (*n* = 2). The primary causes of arrest were significantly different between survivors and non-survivors (*p* = 0.01). Respiratory arrest was the most common cause among survivors, and among non-survivors, “other” was the most common, followed by respiratory causes. Among non-survivors, 59.4% (19 patients) died from the primary cause and cardiopulmonary failure. About 40% (13 patients) who survived the primary cause of cardiac arrest eventually died of septic shock and multiorgan failure.

About 25% of all patients had shockable arrhythmia, and 50% of all patients had seizure attacks within 24 h. The incidence of seizure within 24 h was significantly higher in survivors than in non-survivors. (*p* = 0.006).

The median duration of CPR was 15 min; the duration was significantly longer in non-survivors than the duration for survivors (18 min vs. 5 min, *p* < 0.001). The median NSE within 24 h of cardiac arrest was 43.2 (IQR 31.0–56.5) ng/mL, and the initial lactate level was 10.7 (IQR 4.9–15.7) mmol/L. There was no significant difference in either between survivors and non-survivors.

The use of antiepileptic drugs was significantly higher in survivors than in non-survivors (*p* = 0.001). Extracorporeal membrane oxygenation (ECMO) was applied to only 2 patients. Electroencephalograms (EEG) were performed on 38 (67.8%) patients within 24 h; 21 patients (87.5%) in survivors and 17 patients (53.1%) in non-survivors (*p* = 0.006). The EEG findings are described in Table 1. There was a significant difference between survivors and non-survivors (*p* = 0.003).

### 3.2. EEG Findings after Cardiac Arrest

Normal EEG findings were seen in 2 patients, a burst and suppression pattern was shown in 7 patients (18.4%), diffuse slow delta waves were seen in 21 patients (55.3%), and near-flat/flat EEG was shown in 8 patients (21.1%). The EEG findings were significantly different between survivors and non-survivors (*p* = 0.003). In survivors, diffuse slow delta waves were the most commonly seen patterns (66.7%), and in non-survivors, near-flat or flat was the most commonly seen EEG pattern (47.1%).

### 3.3. Neurologic Outcome after Cardiac Arrest: GOS-E Peds Score

The GOS-E Peds scores were significantly different between survivors and non-survivors (*p* < 0.001). Among survivors, 6 patients (25%) were in score 1–3, 7 patients (29.1%) were in score 4–6, and 11 patients (45.8%) were in score 7. Among non-survivors, 32 patients (100%) were in score 8.

### 3.4. Comparison of Absolute Attenuation and GWR among Survivor, Non-Survivor and Controls

Absolute attenuation in Hounsfield units and GWRs are shown in Table 2. The median ROI of the caudate nucleus, thalamus, medial cortex 1, and medial white matter 1 was significantly lower in the non-survivor group compared to survivors and healthy controls (*p* < 0.001, *p* = 0.003, *p* < 0.001, and *p* = 0.007, respectively).

The median ROI of putamen was 30.0 (IQR: 26.0–32.0), and they were significantly lower in both survivors and non-survivors compared to controls (*p* < 0.001). GWR in basal ganglia was significantly lower in both survivors and non-survivors compared to controls (*p* < 0.001). The median GWR in the cerebrum was higher in non-survivors compared to survivors and controls (*p* = 0.008). Average GWR in both survivors and non-survivors was lower than controls (*p* < 0.001) (Figure 3).

### 3.5. Logistic Regression of Absolute Attenuation in Hounsfield Units and Grey/White Matter Ratio According to Survival

The logistic regression was performed with the dependent variable of survival in the post-cardiac arrest patients. For independent variables, we chose those variables which were significantly different between survivors and non-survivors (*p* < 0.05) (Table 2). The ROI of the caudate nucleus, medial cortex 1, medial white matter 1, medial cortex 2, GWR-cerebrum, and average GWR were included for the logistic regression. The odds ratio of medial cortex 1 was 0.8 (95% CI: 0.67–0.95, *p* = 0.011) and average GWR was 0.0006 (95% CI: 0.0–0.53, *p* = 0.032) (Table 3).

### 3.6. Logistic Regression of Absolute Attenuation in Hounsfield Units and Grey/White Matter Ratio According to Neurologic Outcome

The logistic regression was performed with the dependent variable of neurologic outcome in the post-cardiac arrest patients. For independent variables, we chose those variables which were significantly different between survivors and non-survivors (*p* < 0.05) (Table 4). The ROI of the caudate nucleus, medial cortex 1, medial white matter 1, medial cortex 2, medial white matter 2, and GWR-cerebrum were included for the logistic regression. The medial cortex 1 was a statistically significant variable and the odds ratio was 0.8 (95% CI: 0.68–0.99, *p* = 0.043) (Table 5).

### 3.7. ROC Analyses

As shown in Figure 4, the ROC curve of average GWR showed area under curve (AUC) of 0.666 (*p* = 0.035) and medial cortex 1 showed AUC of 0.733 (*p* = 0.002).

## 4. Discussion

This study presents a single institutional retrospective observational study analyzing the clinical characteristics and the attenuation of grey/white matter, including GWR, in early brain CT scans of children after cardiac arrest to explore the association of GWR with survival and neurological outcome.

Among the clinical characteristics, gender did not show any significant difference between survivors and non-survivors. Although our result showed gender differences in the incidence of cardiac arrest in children, previous researchers have suggested that gender is unrelated to survival in post-cardiac arrest pediatric patients [12,13,14,15]. The primary cause of arrest was most commonly asphyxia, respiratory reasons, and other reasons such as near-drowning. Also, the duration of the CPR was significantly longer in non-survivors which is a reasonable result. The use of hypothermia was higher in survivors.

The prevalence of seizure within 24 h and the use of antiepileptic drugs were significantly higher in survivors compared to non-survivors. To predict neurological outcomes in post CPR patients, pupillary light reflexes, GCS, and several scoring systems have been developed [16,17]. EEG and short-latency somatosensory evoked potentials (SSEP) also have been used to predict poor prognosis from anoxic coma as long as they are applied to appropriate patients. In this study, the EEG findings were also significantly different between survivors and non-survivors. While diffuse slow delta waves were the most frequently found in survivors, in non-survivors, both near flat or flat and diffuse slow delta waves were found in similar proportion. However, SSEP and EEG are influenced by sedative and paralytic agents for intensive care and therapeutic hypothermia, thus difficult to apply for accurate methods of early injury stratification within 24 h of ROSC [18,19,20].

Brain CT is the most frequently used investigational tool that can be used immediately after the stabilization of vital signs. CT findings of the hypoxic damaged brain after cardiac arrest show a global decrease in the cortical gray-matter density from edema, causing loss of the normal gray-white matter differentiation [20,21]. Due to the high metabolic requirement for oxygen and glucose to supply a large number of synapses, gray matter injury is predominant rather than white matter structure. This makes gray matter more susceptible to hypoxic-ischemic injury [2]. Among these findings, a loss of differentiation between gray and white matter on brain CT has been reported and studied.

It is difficult to predict early outcomes from anoxic coma based on CT findings because the loss of GM–WM differentiation could be subtle in children [21,22]; however, the association of GWR has been studied in children resuscitated from cardiac arrest. Starling RM et al. [6] reported the association between early brain CT findings and survival from a single center [6]. They analyzed 78 pediatric patients under 18 years of age who survived cardiac arrest and had a head CT within 24 h of ROCS and concluded that the loss of gray-white matter differentiation and basilar cistern effacement and sulcal effacement is associated with poor outcomes after pediatric cardiac arrest. More recently, Yang et al. reported the analyses of 21 patients from a single center [7]. They reported that unfavorable EEG background and low GWR with hyperammonemia are associated with poor neurologic outcomes in children after cardiac arrest. The AUC of GWR alone was 0.776 in this study, and the combination of EEG, GWR, and ammonia was 0.959 [7]. Tetshuhara et al. retrospectively reviewed 70 patients from a single center and further analyzed 14 survivors’ brain damage. Of note, they used the modified Alberta Stroke Program Early CT Score with GWR. They explored whether a low score was associated with unfavorable neurologic outcomes and revealed the linear correlation of GWR and the pediatric cerebral performance category scale [8]. These studies suggest that the GWR may have an association with survival and neurologic outcome. Although, in adults, large, extended studies have provided a more precise description of GWR for reliable, quantifiable comparisons to predict poor outcomes after hypoxic encephalopathy [4,20,23], predictors of neurologic outcome after CPR have not yet been clearly established in pediatric patients.

In our study, we analyzed each attenuation of gray and white matter then compared them by survival and neurologic outcome. The absolute attenuation of the caudate nucleus, medial cortex, and medial white matter was significantly different between survivors and non-survivors. Also, the GWR of the cerebrum and average GWR were significantly different between survivors and non-survivors. Similarly, the absolute attenuation of the caudate nucleus, medial cortex, and medial white matter was significantly different between patients with better neurologic outcomes (GOS 1–4) and with worse neurologic outcomes (GOS 5–8). The GWR of the cerebrum was significantly low in survivors and patients with better neurologic outcomes. Choi et al. [4] evaluated GWR in 28 adult patients and also revealed decreases in absolute attenuation of the caudate nucleus and putamen in patients who were resuscitated from cardiac arrest compared to controls. Although we could not find any difference in GWR regarding the neurologic outcome, we revealed that the decreased attenuation of the caudate nucleus and both medial cortex and medial white matter in post-CPR pediatric patients were associated with poor neurologic outcome (GOS-E Peds 5–8).

Moreover, the medial cortex at the centrum semiovale level and the average GWR (which is the average of GWR in basal ganglia and cerebrum) were significantly correlated to survival. However, both the medial cortex and GWR had high sensitivity but low specificity. The medial cortex at the centrum semiovale level was also correlated to neurologic outcome significantly. However, it is too early to determine the clinical significance of the medial cortex and GWR as the population is too small.

This study has limitations. The sample size is small, and the number of patients with better neurologic outcomes was small. Also, as the wide range of age and characteristics of the pediatric population, the presented cohort is considerably heterogeneous in terms of the cause of arrest, duration of CPR, post-arrest complications, time to initial CT scan, and other factors. Although GWR was significantly lower in non-survivors compared to survivors, we could not find a correlation with neurologic outcomes, probably due to the small number of patients in GOS-E Peds 1–4. Also, age (0–18 yrs) could show different characteristics in attenuation of brain CT. In addition, the median age of the controls were higher than the study groups (10.5 years vs 5.5 years). A larger sample size would be needed to overcome this limitation. A multi-institutional registry that includes a variety of centers is needed to confirm the predictive value of GWR in neurologic outcomes of pediatric patients.

In conclusion, there was some association of GWR and medial cortex at the centrum semiovale level in early brain CT within 24 h after cardiac arrest with survival. Although we could not find the predictive value of GWR in neurologic outcomes of pediatric patients, the absolute attenuation of the medial cortex may have an association with worse neurologic outcomes. Further prospective, multicenter studies are needed to determine the predictive value of GWR and medial cortex.

## Figures and Tables

**Figure 1 children-09-00036-f001:**
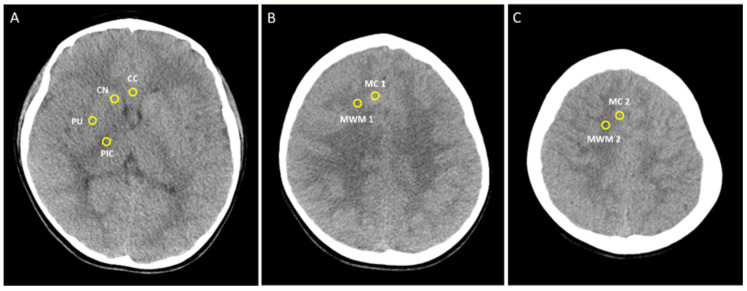
The points of measurement in Hounsfield units of brain computed tomography at (**A**) Basal ganglia level, (**B**) Centrum semiovale level, (**C**) High convexity level. (**A**) Regions of interest (ROI) (10 mm^2^) are positioned in 5 points of the genu of the corpus callosum (CC) (1), caudate nucleus (CN) (2), putamen (PU) (3), posterior lime of internal capsule (PIC) (4). (**B**,**C**) ROI are positioned in 2 points of the medial cortex (MC1 and MC2) and medial white matter (MWM1 and MWM2). Grey and White matter ratio (GWR) in basal ganglia was calculated as follows: (CN + PU)/(CC + PIC). GWR in medial cortex and white matter was calculated as follows: (MC1 + MC2)/(MWM1 + MWM2). Average GWR was defined as the average of these 2 GWR (GWR of basal ganglia + GWR of cerebrum/2).

**Figure 2 children-09-00036-f002:**
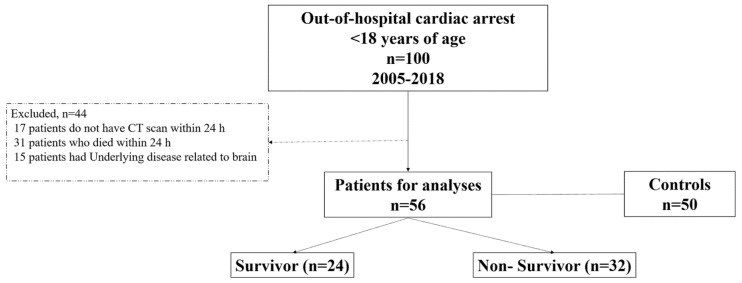
The flowchart of the patients.

**Figure 3 children-09-00036-f003:**
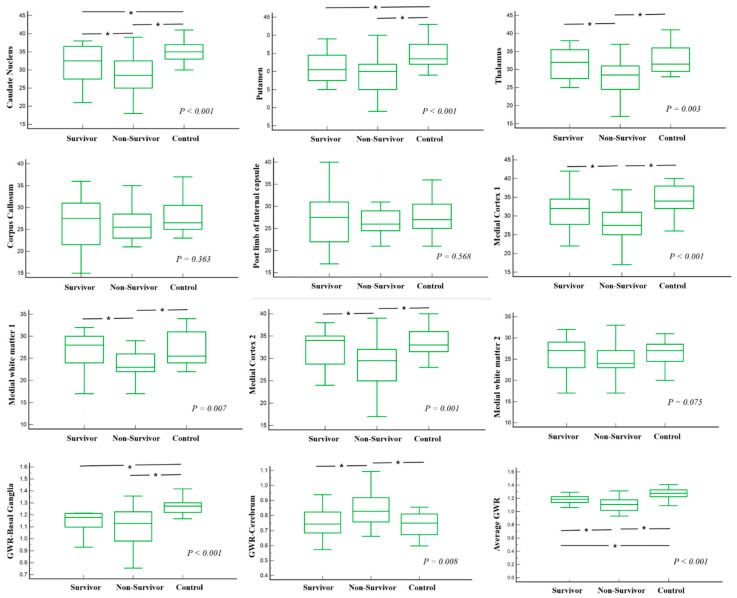
Comparison of absolute attenuations and GWRs among survivors, non-survivors, and healthy controls. * *p* < 0.05.

**Figure 4 children-09-00036-f004:**
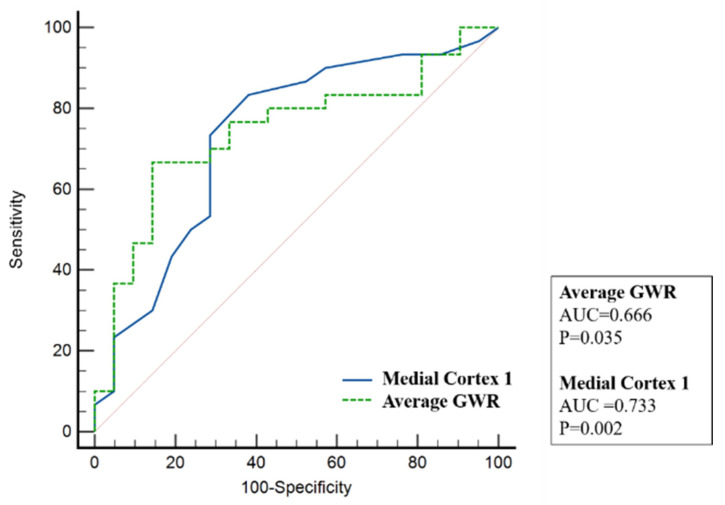
ROC curves to predict survival. The AUC for average GWR was 0.666 (*p* = 0.035) and AUC of medial cortex 1 was 0.733 (*p* = 0.002). ROC, receiver operating characteristic; AUC, area under curve; GWR, grey matter to white matter attenuation ratio.

**Table 1 children-09-00036-t001:** Profiles of study population.

Characteristics	Total (*n* = 56)	Survivor (*n* = 24)	Non-Survivor (*n* = 32)	*p* Value
Age (yrs)	5.5 (0–13)	5.5 (0.5–13)	2.5 (1–8.5)	0.555
Female, *n* (%)	18 (32.1)	7 (29.2)	11 (34.4)	0.197
Primary cause of arrest				0.010
Cardiac, *n* (%)	13 (23.2)	8 (33.3)	5 (15.6)	
Respiratory, *n* (%)	28 (50.0)	15 (62.5)	13 (40.6)	
Other, *n* (%)	15 (26.8)	1 (4.2)	14 (43.6)	
Shockable arrhythmia, *n* (%)	14 (25.0)	8 (33.3)	6 (18.8)	0.216
Seizure, *n* (%)	28 (50.0)	17 (70.8)	11 (34.4)	0.006
Sepsis, *n* (%)	4 (7.1)	1 (4.2)	3 (9.4)	0.099
Duration of CPR (min)	15 (5–28.3)	5 (2.0–10.0)	18 (13.7–30)	<0.001
CT after ROCS (hr)	3 (1–12.2)	5 (1.3–20.7)	2(1–5.5)	0.141
Length of ICU stay (day)	10 (3–30)	13.5 (9.0–24.5)	8 (3.0–22.6)	0.174
Initial NSE (ng/mL)	43.2 (31.0–56.5)	37.8 (26.4–51.3)	43.2 (38.2–109.9)	0.241
Lactate (mmol/L)	10.7 (4.9–15.7)	9.4 (4.9–11.9)	12.3 (4.2–17.2)	0.138
Antiepileptics, *n* (%)	33 (58.9)	20 (83.3)	13 (40.6)	0.001
Use of ECMO, *n* (%)	2 (3.6)	1 (4.2)	1 (3.1)	0.836
EEG done, *n* (%)	38 (67.8)	21(87.5)	17 (53.1)	0.006
EEG finding				0.003
Normal, *n* (%)	2 (5.3)	1 (4.8)	1 (5.9)	
Spikes, *n* (%)	7 (18.4)	6 (28.6)	1 (5.9)	
Delta waves, *n* (%)	21 (55.3)	14 (66.7)	7 (41.2)	
Near flat or flat, *n* (%)	8 (21.1)	0 (0)	8 (47.1)	
Glasgow coma scale	3 (3–4.5)	3 (3–7.2)	3 (3–4)	0.428
GOS-E Peds score				<0.001
GOS-E Peds 1–3	6 (10.7%)	6 (25.0)	0 (0)	
GOS-E Peds 4–6	7 (12.5%)	7 (29.1)	0 (0)	
GOS-E Peds 7–8	43(76.7%)	11 (45.8)	32 (100)	

Abbreviation: ICU, intensive care unit; CPR, cardiopulmonary resuscitation; CT, computed tomography; ECMO, extracorporeal membrane oxygenation; EEG, electroencephalogram; GCS, Glasgow coma scale; GOS-E Peds, Glasgow outcome scale-extended pediatrics; NSE, neuron-specific enolase; ROSC, return of spontaneous circulation.

**Table 2 children-09-00036-t002:** Comparison of absolute attenuation in Hounsfield units and grey/white matter ratio between survivors and non-survivors.

	Total (*n* = 56)	Survivors (*n* = 24)	Non Survivors (*n* = 32)	*p* Value
Caudate nucleus	32.0 (27.5–36.0)	32.5 (27.5–36.5)	28.5 (25.0–32.5)	0.014
Putamen	31.0 (28.0–34.5)	30.5 (27.5–34.5)	30.0 (25.0–32.0)	0.288
Corpus callosum	26.0 (23.5–31.0)	27.5 (21.5–31.0)	25.5 (23.0–28.5)	0.652
Post limb of internal capsule	27.0 (24.0–30.0)	27.5 (22.0–31.0)	26.0 (24.5–29.0)	0.535
Medial cortex1	31.0 (26.2–34.0)	32.0 (27.7–34.5)	27.5 (25.0–31.0)	0.004
Medial white matter 1	25.0 (22.2–28.7)	28.0 (24.0–30.0)	23.0 (22.0–26.0)	0.005
Medial cortex2	32.0 (28.0–35.0)	34.0 (28.7–35.0)	29.5 (25.0–32.0)	0.012
Medial white matter 2	26.0 (23.0–28.0)	27.0 (23.0–29.0)	24.0 (23.0–27.0)	0.104
GWR in Basal ganglia	1.19 (1.09–1.27)	1.17 (1.09–1.21)	1.12 (0.98–1.22)	0.187
GWR in cerebrum	0.78 (0.70–0.85)	0.74 (0.68–0.82)	0.83 (0.75–0.91)	0.026
Average GWR	1.18 (1.08–1.26)	1.18 (1.13–1.22)	1.10 (1.01–1.17)	0.034

**Table 3 children-09-00036-t003:** Logistic regression of absolute attenuation in Hounsfield units and grey/white matter ratio.

	Odds Ratio	95% CI	*p*-Value
Caudate nucleus	1.11	0.85–1.44	0.415
Medial cortex 1	0.80	0.67–0.95	0.011
Medial white matter 1	0.68	0.41–1.13	0.138
Medial cortex 2	1.04	0.72–1.52	0.805
GWR in cerebrum	3.35	0.00–35.4	0.774
Average GWR	0.00	0.00–0.53	0.032

**Table 4 children-09-00036-t004:** Comparison of absolute attenuation in Hounsfield units and grey/white matter ratio between better neurologic outcome and poorer neurologic outcome.

	GOS 1–4 (*n* = 6)	GOS 5–8 (*n* = 50)	*p*-Value
Caudate nucleus	36.0 (31.0–37.0)	30.0 (26.0–33.0)	0.049
Putamen	33.5 (30.0–37.0)	30.0 (26.0–32.0)	0.177
Corpus callosum	31.0 (29.0–31.0)	25.5 (23.0–29.0)	0.114
Post limb of internal capsule	30.5 (26.0–31.0)	26.0 (23.0–29.0)	0.117
Medial cortex1	33.5 (32.0–36.0)	29.0 (25.0–32.0)	0.013
Medial white matter 1	29.0 (28.0–30.0)	24.0 (22.0–27.0)	0.009
Medial cortex2	35.0 (35.0–36.0)	30.0 (25.7–33.0)	0.002
Medial white matter 2	28.0 (27.0–29.0)	24.0 (22.7–27.0)	0.018
GWR in Basal ganglia	1.14 (1.10–1.20)	1.13 (1.02–1.22)	0.851
GWR in cerebrum	0.70 (0.68–0.75)	0.80 (0.74–0.90)	0.049
Average GWR	1.20 (1.17–1.22)	1.13 (1.02–1.19)	0.063

**Table 5 children-09-00036-t005:** Logistic Regression of Absolute Attenuation in Hounsfield Units and Grey/White Matter Ratio According to Neurologic Outcome.

	Odds Ratio	95% CI	*p*
Cudate nucleus	0.99	0.81–1.22	0.976
Medial cortex 1	0.83	0.68–0.99	0.043
Medial white matter 1	0.96	0.65–1.41	0.831
Medial cortex 2	0.93	0.64–1.34	0.699
Medial white matter 2	1.14	0.74–1.74	0.542
GWR in cerebrum	14.7	0.00–34.7	0.496

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
