# Peer review of "Clinical Significance of Gray to White Matter Ratio after Cardiopulmonary Resuscitation in Children"

_children, 2022, doi:10.3390/children9010036_

Round 1
Reviewer 1 Report
Thank you for the opportunity to review this manuscript.
The authors present a single institutional retrospective study analyzing the grey to white matter ratio (GWR) in early brain CT scans of children after cardiac arrest to explore association of GWR with neurological outcome. They included 72 children across the entire pediatric age spectrum. From a clinical point of view, the subject and intention of this study is certainly of relevance and interest for the community of paediatric instensivists and all specialties involved in the post-rescucitation care and diagnostics of children. As the authors correctly state, prognostication of neurologic outcome after cardiac arrest, already quite challenging in adult patients, is still often not possible with sufficient diagnostic certainty. It is quite sure, that no single diagnostic modality will ever allow such prognostication with adequate accuracy. Nonetheless it is important to gather more data to allow a better evaluation of each modality’s diagnostic capacities and pitfalls.
The study I well and concisely written. Inclusion and exclusion criteria are reasonable. The sample size is somewhat limited, as it is frequently the case in pediatric studies, however, cardiac arrests in children are fortunately rather rare.
However, there are several major and minor concerns regarding study design, data presentation and discussion. In the present form, I cannot recommend publication but would advice a thorough revision of the study and manuscript instead, addressing the following concerns:
Methodology
-I suppose written consent from resuscitated children and healthy controls was not obtained. Please state if consent was waived by ethic committee
-It is stated that GWR was assessed by two independent observers to rate inter-observer agreement. How was data integrated? How was inter-observer variability, which is an important issue, assessed?
-were observers analyzing CT data blinded for clinical outcome? How was blinding guaranteed? If not, how was bias ruled out?
-How was data distribution assessed? I assume that most of the data presented does not have a normal distribution. Therefore median and interquartile range is the adequate format of presentation. Also non-parameteric tests will then have to be used.
-The univariable and multivariable analyses lack any meaningful description. How was data handled, how was it entered (continuously, dichotomized?), what regression was used? How was decided about inclusion/exclusion from multivariable model? How were models compared? Methodology must be adequately described. Also results should be presented in a table.
-Is there a standard of care for post-resuscitation care in children? What were therapy goals. Were seizures diagnosed / treated? They can be an important factor that may trigger secondary brain injury. Were there any cases with ECMO? Sepsis is also known to be associated with poor neurologic outcome in critically ill pediatric patients, in what proportion was infection present and treated?
-regarding The GWR assessment / calculation, several somewhat differing approaches and ROIs have been described previously. How are methodological differences of relevance? How was the author’s way chosen? How could that influence comparability of data? Please explain and discuss.
Results
-there is a disparity in gender with 32:68%. Is there any explanation or assumption? Please discuss.
-Importantly, no mode and cause of death are presented. How and when did non-surviving patients die? Did any results of brain imaging influence therapeutic decisions (e.g. withdrawal of support). Can a “self-fulfilling prophecy be ruled out?
- were there any other diagnostics performed (SSEP, EEG, biomarkers) etc?
- the grouping of GOS-E Peds 1-6 included patients with good recovery as well as patients with “lower severe disability”. I Think most pediatricians and pediatric neurologists would agree that severe disability might not be considered a “good” outcome after resuscitation. The aouthors might want to re-think their classification/grouping. Or discuss this matter and their reasons to group patients in that manner.
Discussion:
The discussion is in general quite superficial and does not adequately discuss available data from the literature. In fact, the referenced studies are comparably old (newest 2013). No previous data from children was discussed.
- There are numerous more recent studies of GWR in adults, that also indicate various possible confounding factors (such as time of CT after arrest, gender, age among others) and in general show a good specificity (with varying cut-offs) with a rather poor specificity for GWR to predict poor outcome. The findings of the present study will have to be meaningfully discussed in respect to these data and results. Otherwise the presented findings are hardly interpretable for readers less familiar with the topic.
- Importantly, there are also various previous studies that have assessed GWR in children after CA, also from Korea where the author’s study was conducted(e.g. Starling et al., Pediatr Crit Care Med . 2015 Jul;16(6):542-8. doi: 10.1097/PCC.0000000000000404; Donghwa et al., Front Pediatr. 2019 Jun 4;7:223. doi: 10.3389/fped.2019.00223.; Tetsuhara et al., Sci Rep. 2021 Jun 8;11(1):12090. doi: 10.1038/s41598-021-91628-y) These references and previous, and probably more, will have to be referenced and discussed adequately. Especially in pediatrics, age-dependent affects are important. There is previous data, that cut-offs for GWR might differ according to age (see references above). This has not been considered in the present study.
-It was stated that “Regarding the importance of thalamus, modified pediatric calculation method of GWR including thalamus can be helpful.” (line 203f) What is that supposed to mean?
Author Response
Response to Reviewer 1
Thank you very much for your valuable inputs to our manuscript. We all agreed to your critical point and amended according to your comments. We did not have enough time for English editing service, please bear with my English. If you give us the chance, we will edit the English additionally. Again, thank you very much for your time and input. Here are our point by point responses.
The authors present a single institutional retrospective study analyzing the grey to white matter ratio (GWR) in early brain CT scans of children after cardiac arrest to explore association of GWR with neurological outcome. They included 72 children across the entire pediatric age spectrum. From a clinical point of view, the subject and intention of this study is certainly of relevance and interest for the community of paediatric instensivists and all specialties involved in the post-rescucitation care and diagnostics of children. As the authors correctly state, prognostication of neurologic outcome after cardiac arrest, already quite challenging in adult patients, is still often not possible with sufficient diagnostic certainty. It is quite sure, that no single diagnostic modality will ever allow such prognostication with adequate accuracy. Nonetheless it is important to gather more data to allow a better evaluation of each modality’s diagnostic capacities and pitfalls.
The study I well and concisely written. Inclusion and exclusion criteria are reasonable. The sample size is somewhat limited, as it is frequently the case in pediatric studies, however, cardiac arrests in children are fortunately rather rare.
However, there are several major and minor concerns regarding study design, data presentation and discussion. In the present form, I cannot recommend publication but would advice a thorough revision of the study and manuscript instead, addressing the following concerns:
Methodology
-I suppose written consent from resuscitated children and healthy controls was not obtained. Please state if consent was waived by ethic committee
Yes. As the study design was retrospective, the written conset were waived by ethics committee.
On Page 2, we added “the written consents were waived by ethics committee.”
-It is stated that GWR was assessed by two independent observers to rate inter-observer agreement. How was data integrated? How was inter-observer variability, which is an important issue, assessed?
Thank you for your critical point. Before assessment, both had several meeting to more accurately measure with minimal inter-observer variability. The inter-observer variability was minimal as we discussed the points we will measure and also the hardware system was same. We put the mean of both values. (a+b/2) We also added on page 2, “We checked the inter-observer variability and then used the mean of the value.”
-were observers analyzing CT data blinded for clinical outcome? How was blinding guaranteed? If not, how was bias ruled out?
The CT data observers were totally blinded for clinical outcome, because the CT data and outcome data were separately and simultaneously measured. No one in the group did not have any clue of the outcomes. We added this on page 2, “For inter-rater agreement in statistics, ROIs were determined independently by two raters and both were blinded to the clinical outcome.”
-How was data distribution assessed? I assume that most of the data presented does not have a normal distribution. Therefore median and interquartile range is the adequate format of presentation. Also non-parameteric tests will then have to be used.
This is also a very important point you gave us. We changed all the valued to median and IQR.
-The univariable and multivariable analyses lack any meaningful description. How was data handled, how was it entered (continuously, dichotomized?), what regression was used? How was decided about inclusion/exclusion from multivariable model? How were models compared? Methodology must be adequately described. Also results should be presented in a table.
We have reanalyzed all the statistics again. The previous analyses did not contain univariate and multivariate analyses. On page 3, we described the statistical analyses in more detail.
“Continuous variables are expressed as median and Interquartile range (IQR). The Mann–Whitney test were used to compare continuous variables between 2 groups (survivors vs. Non survivors) and Kruskal-Wallis test for non-normally distributed data were used to compare continuous variables among three groups (Survivors vs. Non survivors vs. Healthy controls). A Chi-square test was used to compare non-continuous variables. For post-hoc ana-lyses, the Tukey method was used. In all analyses, p-values < 0.05 were considered to be statistically significant. Logistic regres-sion was performed in those which were statistically significant. ROC analyses were performed. All analyses were performed using MedCalc Statistical Software ver. 19.1 (MedCalc Software bvba; Ostend,Belgium;http://www.medcalc.org; 2019)”
-Is there a standard of care for post-resuscitation care in children? What were therapy goals. Were seizures diagnosed / treated? They can be an important factor that may trigger secondary brain injury. Were there any cases with ECMO? Sepsis is also known to be associated with poor neurologic outcome in critically ill pediatric patients, in what proportion was infection present and treated?
Thank you for your critical comments. The ultimate goal is the patient has the minimal or no brain injury and be back to normal life. To do so, evaluation of the brain, and control seizure, protect the brain while protecting other vulnerable organs are our standard of care for post-resuscitation care. As soon as the patients has ROCS, they take quick brain CT on the way to PICU. (We have CT room on the way from ER to ICU) We recollected data for the incidence of seizure, the use of antiepileptic drugs, ECMO and sepsis and analyzed. We added further variables in Table 1.
-regarding The GWR assessment / calculation, several somewhat differing approaches and ROIs have been described previously. How are methodological differences of relevance? How was the author’s way chosen? How could that influence comparability of data? Please explain and discuss.
We appreciated your comments. The calculation of GWR was done in a traditional way but the reason for the additional circle of ROI was for Thalamus. We know thalamus is important basal ganglia especially in children, so out of curiosity, we checked thalamus additionally. The absolute attenuation of thalamus was significantly lower in non survivors compared to survivors and controls. (as described Table 2) However, we agree, adding thalamus in the figure 1 might confuse readers, so deleted from the figure 1.
“ROIs were assessed bilaterally in caudate nucleus (CN), putamen (PU), forceps minor of the corpus callosum (CC), posterior limb of internal capsule (PIC), medial cortex and medial white matter at the level of the centrum semiovale and high convexity (MC1/MC2, MWM1/MWM2). GWR in basal ganglia was calculated as followings: (CN + PU) / (CC + PIC). GWR in medial cortex and white matter was calculated as following: (MC1 + MC2) / (MWM1 + MWM2). Average GWR was defined as the average of these 2 GWR (GWR of basal ganglia + GWR of cerebrum/2) [5].”
Results
-there is a disparity in gender with 32:68%. Is there any explanation or assumption? Please discuss.
Thank you for your comments. Yes. There was sexual disparity in this cohort. Among the clinical characteristics, the gender did not show any significant difference between survivors and non survivors. However, the male sex tends to be higher than female sex. The boys tend to be more active in water sports and outdoor activities which matches to the primary cause of arrest in this study. The primary cause of arrest was most commonly asphyxia, respiratory reason and other reasons such as near drowning. We added this in the discussion session.
-Importantly, no mode and cause of death are presented. How and when did non-surviving patients die? Did any results of brain imaging influence therapeutic decisions (e.g. withdrawal of support). Can a “self-fulfilling prophecy be ruled out?
Thank you for your critical comment. The non surviving patients died of hemodynamic instability, infections, however with the data we have at the moment, we could not run any statistics. We will have the data available and prepare for the next revision.
- were there any other diagnostics performed (SSEP, EEG, biomarkers) etc?
Yes. There were results of EEG and neuron specific enolase. We added these to the results and table. We did not perform SSEP in this cohort.
- the grouping of GOS-E Peds 1-6 included patients with good recovery as well as patients with “lower severe disability”. I Think most pediatricians and pediatric neurologists would agree that severe disability might not be considered a “good” outcome after resuscitation. The aouthors might want to re-think their classification/grouping. Or discuss this matter and their reasons to group patients in that manner.
Thank you for pointing out this issue. This was the most painful part and we agree that most pediatricians would not agree with the classification. However, as the number of better outcomes (GOS 1-3) were too low, while GOS 7-8 were very high, we had decided to divide so and analysed accordingly. This time, we wanted to re-think classification and divided the patients into survivors and non-survivors. But the neurologic outcome was still important and curious part of ours, so we added comparison between patients with better neurologic outcome and worse outcome: GOS 1-4 (n=6) vs GOS 5-8 (n=50). Although the numbers seem very deviated, we should look forward to further study in near future. Please give us any flank opinion of yours. We are more than happy to revise again.
Discussion:
The discussion is in general quite superficial and does not adequately discuss available data from the literature. In fact, the referenced studies are comparably old (newest 2013). No previous data from children was discussed.- There are numerous more recent studies of GWR in adults, that also indicate various possible confounding factors (such as time of CT after arrest, gender, age among others) and in general show a good specificity (with varying cut-offs) with a rather poor specificity for GWR to predict poor outcome. The findings of the present study will have to be meaningfully discussed in respect to these data and results. Otherwise the presented findings are hardly interpretable for readers less familiar with the topic.
Yes. We agree the discussion was inadequately discussion the data. We have added our results and discussed in the discussion session. Also, we added the new citations to our manuscript.
- Importantly, there are also various previous studies that have assessed GWR in children after CA, also from Korea where the author’s study was conducted(e.g. Starling et al., Pediatr Crit Care Med . 2015 Jul;16(6):542-8. doi: 10.1097/PCC.0000000000000404; Donghwa et al., Front Pediatr. 2019 Jun 4;7:223. doi: 10.3389/fped.2019.00223.; Tetsuhara et al., Sci Rep. 2021 Jun 8;11(1):12090. doi: 10.1038/s41598-021-91628-y) These references and previous, and probably more, will have to be referenced and discussed adequately. Especially in pediatrics, age-dependent affects are important. There is previous data, that cut-offs for GWR might differ according to age (see references above). This has not been considered in the present study.
Thank you. We have added the pediatric papers in the discussion session.
-It was stated that “Regarding the importance of thalamus, modified pediatric calculation method of GWR including thalamus can be helpful.” (line 203f) What is that supposed to mean?
We have deleted this sentence.

Reviewer 2 Report
I am glad to read the article entitling "Clinical significance of gray to white matter ratio after cardio-pulmonary resuscitation in children" written by Yun-Young Lee et al.
In general, this paper was well-written, even though with a relatively negative results "no difference of GWR regarding neurologic good or impaired groups, but with absolute attenuation of gray matter in post-CPR patients."
Comments:
- CPR-to-CT duration is relatively long (9.33 hour+/- 9,29), which may influence the gray matter image density secondary to progressive brain tissue edema and anoxic change. Is there any CT number changes with regard of CPR-to-CT duration?
- As understood, neurologic outcome after CPR is influenced by many factors in addition to initial brain ischemic insults (eg. post-CPR vital sign, post-CPR hypothermia, underlying diseases.) Therefore, it is not surprising to the reviewer that mean value by t-test does not show a significant difference in good and poor neurological patient groups. Is the any cut-off value with a poor prognostic prediction? By which you can suggest a initial CT criteria with a poor neurological outcome.
- Suggest a flowchart for this study. How many cases include/exclude in the first place, how many cases were excluded because they can not survive over 24 hours, how many cases became consciousness clear within 6 hours, ... . I think this flowchart can help readers to read and interpret the study manuscript more clear and precise.
Author Response
Response to Reviewer 2
Thank you very much for your valuable inputs to our manuscript. We all agreed to your critical point and amended according to your comments. Here are our point by point responses.
- CPR-to-CT duration is relatively long (9.33 hour+/- 9,29), which may influence the gray matter image density secondary to progressive brain tissue edema and anoxic change. Is there any CT number changes with regard of CPR-to-CT duration?
Thank you very much for your critical comments. Some of data did not have a normal distribution and we expressed the data with median (IQR) with non-parametric tests.
In Table 1, the ROSC to CT duration was median 3 hours (1-12.2h). In survivors, the median duration was 5 hours (IQR: 1.3-20.7h) and in non-survivors, the median duration was 2 hours (IQR: 1-5.5h). Some extreme values must have resulted the mean value longer than it is. There was no statistical difference between survivors and non survivors regarding the ROSC to CT duration. Please let us know if we need additional analyses, we are more than happy to follow your instruction.
- As understood, neurologic outcome after CPR is influenced by many factors in addition to initial brain ischemic insults (eg. post-CPR vital sign, post-CPR hypothermia, underlying diseases.) Therefore, it is not surprising to the reviewer that mean value by t-test does not show a significant difference in good and poor neurological patient groups. Is the any cut-off value with a poor prognostic prediction? By which you can suggest a initial CT criteria with a poor neurological outcome.
Thank you. We agree there are very many factors incluencing the neurologic outcome. The classification was the most painful part, as the number of better outcomes (GOS 1-3) were too low, while GOS 7-8 were very high. This time, we wanted to re-think classification and divided the patients into survivors and non-survivors. But the neurologic outcome was still important and curious part of ours, so we added comparison between patients with better neurologic outcome and worse outcome: GOS 1-4 (n=6) vs GOS 5-8 (n=50). Although the numbers seem very deviated, we should look forward to further study in near future.
In table 1, we added more variables such as, incidence of seizure, sepsis, level of NSE, use of ECMO, use of hypothermia and EEG findings and found, the incidence of seizure, duration of CPR, use of hypothermia and EEG findings were significantly different between survivors and non survivors.
We also analysed the logistic regression of absolute attenuation in Hounsfield units and grey/white matter ratio (Table 2 and 3), and found the medial cortex 1 and average GWR were significant factors. We added the ROC analyses to the result with these 2 variables. (Figure 4)
Please give us any flank opinion of yours. We are more than happy to revise again.
- Suggest a flowchart for this study. How many cases include/exclude in the first place, how many cases were excluded because they can not survive over 24 hours, how many cases became consciousness clear within 6 hours, ... . I think this flowchart can help readers to read and interpret the study manuscript more clear and precise.
We appreciate your suggestion. We believe the flowchart will help readers to interpret the study better. We added the flowchart in the manuscript (Figure 2)
Thank you again for your time to read and make it better. Thank you.

Round 2
Reviewer 1 Report
The authors have substantially revised the manuscript and several of my concerns were addressed. Overall, the manuscript has considerably improved in my point of view, however, there remain some points that will require additional revision
General
-the manuscript now requires extensive language editing
Methodology
-It is now stated that inter-observer agreement was checked when assessing GWR. Please specify inter-observer variability and how the authors chose to mean results of both investigators and how their methodology might influence study results
Results / discussion
-Concerning the cohort’s gender disparity the authors state “The boys tend to be more active in water sports and outdoor activities which matches to the primary cause of arrest in this study”. Such statements seem rather stereotypical and are most likely not grounded on any substantial data. Please provide data or references that support these statements (gender disparities in the incidence of cardiac arrest/ arrest causes in the author’s country? Etc) or clearly mark these as pure speculations or, preferably, remove such statements if unsubstantiated.
-Still, mode and cause of death are presented. (see my previous comment)
-The discussion has improved but still does hardly mention relevant results from previous studies. The discussion is intended to allow readers a comparison and facilitate a meaningful interpretation of the results. In this respect it will be helpful to present some specific data from previous studies. Were attenuations comparable in magnitude and localization, or did the findings differ relevantly? Were there cut-offs suggested previously? Are there methological differences to previous studies? What makes the authors’ study relevant in comparison to previous studies? These aspects should be adequately included in the discussion section.
-The presented cohort is considerably heterogeneous in terms of cause of arrest, duration of CPR, post-arrest complications (seizures, sepsis…), time point at which CT scan was performed, age and probably numerous more factors. The small sample size does not allow to control for such confounders which introduces significant uncertainty interpreting results. As such, I think that the studies’ limitations are still insufficiently discussed and more importantly, the results are somewhat overstated and the discussion should be toned down accordingly. I would strongly suggest not to use the term “prediction of outcome” since the author’s results are hardly suitable for outcome prediction. Rather stick to terms like association or with ROC analysis to classification/discrimination. Moreover, how do the authors interprete their results of ROC analysis?
Please also refer to the STROBE Statement—checklist (https://www.equator-network.org/wp-content/uploads/2015/10/STROBE_checklist_v4_combined.pdf) for the recommended reporting of observational studies: 18. Summarise key results with reference to study objectives. 19. Discuss limitations of the study, taking into account sources of potential bias or imprecision. Discuss both direction and magnitude of any potential bias. 20. Give a cautious overall interpretation of results considering objectives, limitations, multiplicity of analyses, results from similar studies, and other relevant evidence. 21. Discuss the generalisability (external validity) of the study results…..
Author Response

(The authors gave the same response as above.)
